# Trends in the distribution of body mass index, waist circumference and prevalence of obesity among Taiwanese adults, 1993–2016

**Tzu-Jung Wong[1,2], Tsung Yu[3]***

**1** Department of Healthcare Information and Management, School of Health Technology, Ming Chuan University, Taoyuan, Taiwan, **2** Department of Academic Clinical Programme, National Dental Centre Singapore, Singapore, Singapore, **3** Department of Public Health, College of Medicine, National Cheng Kung University, Tainan, Taiwan

* tsungyu@mail.ncku.edu.tw

## Abstract

### Background

Differences in the prevalence of general and abdominal obesity by subgroups such as age, sex, and education have been reported worldwide. Most studies in Taiwan regarding obesity prevalence were targeted at school-aged children or without further stratification by subgroups. Our aim was to examine the age-specific secular trend of body mass index (BMI), waist circumference (WC), and obesity prevalence stratified by sex, education and urbanization levels in Taiwanese adults.

### Methods

We used three waves of nationally representative population from the Nutrition and Health Survey in Taiwan (NAHSIT) 1993–1996 (n = 2 989), 2005–2008 (n = 2 495), and 2013–2016 (n = 2 880). The data included standardized measurement of body weight, height, and WC. We conducted a serial cross-sectional analysis among adults aged 20 years or above to examine the age-specific trends of BMI, WC, and the prevalence of underweight, overweight, general obesity, and abdominal obesity with stratification by sex, education, and urbanization levels.

### Results

The general obesity prevalence was 16%, 21%, and 20% and the abdominal obesity prevalence was 27%, 42%, and 47% in the 1993–1996, 2005–2008, and 2013–2016 surveys, respectively. The age-specific secular trend of BMI differed across subgroups; however, the trend of WC increased rapidly regardless of subgroups, except for women aged ≥60 years. The general obesity prevalence increased noticeably among men, younger- and middle-age adults with high school or higher education, middle- and older-age adults with lower than high school education, people <39 and ≥50 years of age residing in rural areas, and among those between 30 and 59 and ≥70 years of age residing in urban areas.

**Data Availability Statement:** Data cannot be shared publicly according to the policy in Taiwan. Data are owned by the Health and Welfare Data Science Center, Ministry of Health and Welfare,

Taiwan (https://dep.mohw.gov.tw/DOS/cp-5119-59201-113.html) and were available for researchers who meet the criteria for access to confidential data.

**Funding:** Ministry of Science and Technology, Taiwan (110-2314-B-006-050). The funders had no role in study design, data collection and analysis, decision to publish, or preparation of the manuscript.

**Competing interests:** The authors have declared that no competing interests exist.

## Conclusions

Although the increasing trend of general obesity prevalence was levelling off among several subgroups, the abdominal obesity prevalence increased significantly and rapidly in Taiwan. Future research in developing effective weight and WC control interventions tailored to different subgroups is urgently needed.

## Introduction

Body mass index (BMI) is an anthropometric measure that is easy to obtain to categorize a person's general adiposity. Although BMI has limitations when it is applied to individuals with abdominal obesity or high muscle mass, it is predictive of morbidity and mortality in the general population [1]. Meta-analysis using individual participant data shows that compared with people with BMI of 20 to 25 kg/m$^2$, people with BMI below this range (underweight) or above this range (overweight) have slightly higher risk of all-cause death and the risk gets increasingly higher for obese people (BMI >30 kg/m$^2$) [2]. These findings are generalizable to many populations worldwide.

However, since the BMI measure does not differentiate lean body mass from fat body mass and it does not reflect the distribution of body fat, researchers have suggested that other anthropometric measures such as waist circumference (WC) can also be used to predict morbidity and mortality [3]. The INTERHEART study, for instance, examined the association of BMI, WC, hip circumference, and waist-to-hip ratio with myocardial infarction; they found that the waist-to-hip ratio was most predictive of myocardial infarction [4]. It is thereby important to consider abdominal adiposity in addition to general adiposity while we are assessing the cardio-metabolic health of a population.

Understanding the secular trends (long-term trends) of general obesity and abdominal obesity would be informative for designing interventions and policies. Population-based research findings regarding the recent trends in the prevalence of obesity vary from country to country. In the US, the National Health and Nutrition Examination Survey showed an increasing linear trend in general obesity from 2005 to 2014 among women but not among men, after accounting for age, race/ethnicity, smoking, and education [5]. In Asia, the China Health and Nutrition Survey showed that from 1993 to 2015 the prevalence of overweight, general obesity and abdominal obesity increased markedly, particularly in men [6]. However, reports from Hong Kong and Korea indicated that trends in the prevalence of overweight/general obesity among women have levelled off or declined in recent years [7,8].

In Taiwan we found only one report discussing the secular trends in the prevalence of overweight and obesity using a population-based survey. Yeh and colleagues compared only two waves of the Nutrition and Health Survey in Taiwan (NAHSIT 1993–1996 and NAHSIT 2005–2008) and found that the prevalence of overweight, general obesity, and metabolic syndrome increased in both men and women [9]. Using the previous surveys and the recent NAHSIT 2013–2016, our goal was to (1) examine the trends in the distribution of BMI and waist circumferences in Taiwan, and (2) examine the trends in the prevalence of underweight, overweight, general obesity, and abdominal obesity in Taiwan. Our analysis of trends was age-specific; we stratified by sex, education and urbanization.

## Methods

The Nutrition and Health Survey in Taiwan (NAHSIT) was a government-sponsored national-representative cross-sectional health examination and nutrition survey conducted by the Ministry of Health and Welfare in Taiwan since 1992. The present study included three

waves of the survey, NAHSIT 1993–1996, 2005–2008, and 2013–2016. The survey team in each wave used multi-staged stratified and clustered sampling to obtain a nationally representative sample of the Taiwanese non-institutionalized population; strata were formed based on the townships/districts in Taiwan and the probability proportional to population size method was used for sampling. The survey consisted of two components: (1) face-to-face household survey that was conducted at the participants' homes, and (2) physical examination that was conducted at a temporary health examination station. The survey details have been reported elsewhere and all procedures of the study have been approved by the responsive ethics boards [10–13]. Datasets were obtained from the Health and Welfare Data Science Center, Ministry of Health and Welfare, Taiwan.

We included participants who were 20 to 79 years of age. There were 4 859, 4 404, and 5 377 participants aged 20 to 79 years in surveys conducted in 1993–1996, 2005–2008, and 2013–2016, respectively. Only about 60% of the participants attended the health examination. We included participants without missing data on weight or height values and excluded pregnant women. In total, our analysis had 62% (n = 2 989) participants of the 1993–1996 survey, 57% (n = 2 495) of the 2005–2008 survey, and 54% (n = 2 880) of the 2013–2016 survey.

Standardized equipment and techniques were used to measure WC in centimeters (cm), weight in kilograms (kg), and height in cm at the temporary health examination station. BMI was calculated as weight in kg divided by height in meters (m) squared. The BMI was categorized into four groups according to the recommendation of the World Health Organization for Asians: underweight ($<18.5$ kg/m$^2$), normal weight (18.5– $<23$ kg/m$^2$), overweight (23– $<27.5$ kg/m$^2$), and obese ($\geq 27.5$ kg/m$^2$). The abdominal obesity was defined as WC of $> 90$ cm for men and $> 80$ cm for women.

The demographic factors used in this study included age, sex, educational level, and urbanization level at interview. Participants were divided into six age groups with a 10-year cut-off: 20 to 29 years, 30 to 39 years, 40 to 49 years, 50 to 59 years, 60 to 69 years, and 70 to 79 years. Sex was categorized as female or male. The educational levels were grouped into two categories: less than high school and high school or above. For urbanization level, Liu in 2006 grouped the residential areas in Taiwan into seven categories (1–7), from the most urbanized to the least urbanized. The urbanization level of residential areas in this study was re-categorized into two groups: urban (categories 1–3) and rural (categories 4–7) areas.

Analyses were conducted separately for BMI, WC, abdominal obesity, general obesity, overweight, and underweight, stratified by sex, educational level, and urbanization level. For each stratification, we conducted age-group-specific analysis. The complex survey design and the differential probabilities of selection that resulted from non-response and non-coverage were taken into account with proper weighting when performing the analyses for the survey samples. Mean and standard error (SE) were reported for the continuous variables: BMI and WC. Simple linear regression was used to examine the secular trends in the mean value of BMI and WC over the three waves of the 4-year survey cycle. For categorical variables, including abdominal obesity, general obesity, overweight, and underweight, the percentage and SE were reported. Simple logistic regressions were used to examine the secular trends in the prevalence of abdominal obesity, general obesity, overweight, and underweight. To determine the statistical significance, a two-sided *P* value of less than 0.05 was considered. All analyses were conducted using Stata version 15 (StataCorp LLC, College Station, TX, USA) with the *svy* commands.

## Results

We summarize the prevalence of the selected characteristics for each wave of samples in Table 1. The distributions of age and sex were similar among survey waves. Men accounted for

**Table 1. Characteristics of the sample.**

|  | 1993–1996 NAHSIT | 2005–2008 NAHSIT | 2013–2016 NAHSIT |
|---|---|---|---|
| N | 2 989 | 2 495 | 2 880 |
| Men (%) | 1 407 (47) | 1 211 (49) | 1 393 (48) |
| Age distribution (%) |  |  |  |
| 20–29 | 328 (11) | 302 (12) | 376 (13) |
| 30–39 | 500 (17) | 269 (11) | 366 (13) |
| 40–49 | 613 (21) | 476 (19) | 403 (14) |
| 50–59 | 609 (20) | 491 (20) | 588 (20) |
| 60–69 | 644 (22) | 467 (19) | 672 (23) |
| 70–79 | 295 (10) | 490 (20) | 475 (16) |
| BMI in kg/m$^2$ (SD) | 24.0 (3.8) | 24.6 (4.0) | 24.5 (4.1) |
| Weight status (%) |  |  |  |
| Underweight | 169 (6) | 98 (4) | 120 (4) |
| Ideal | 1 083 (36) | 814 (33) | 1 026 (36) |
| Overweight | 1 251 (42) | 1 070 (43) | 1 150 (40) |
| Obesity | 486 (16) | 513 (21) | 584 (20) |
| Waist circumference in cm (SD) | 78.9 (10.2) | 83.2 (10.9) | 84.8 (11.4) |
| Abdominal obesity (%) | 816 (27) | 1 057 (42) | 1 355 (47) |
| Attained high school or above education (%) | 842 (28) | 1 162 (47) | 1 757 (61) |
| Urban area (%) | 1 087 (36) | 1 155 (46) | 1 392 (48) |

BMI = body mass index, NAHSIT = Nutrition and Health Survey in Taiwan, SD = standard deviation.

47% (n = 1 407/2 989), 49% (n = 1 211/2 495), and 48% (n = 1 393/2 880) in the 1993–1996, 2005–2008 and 2013–2016 surveys, respectively. During the three survey waves conducted across 20 years, the prevalence of people who attained high school or above increased more than two fold, from 28% to 61%. The prevalence of people in the urban residential areas also increased from 36% to 48%.

Table 2 shows the trends in the BMI and WC mean values for each age group stratified by sex, educational levels and urbanization levels among Taiwanese adults for the three waves of the survey. Overall, people in the 1993–1996 survey had the lowest BMI mean values. Among men, BMI mean values ranged from 21.78 to 23.70 in the 1993–1996 survey, 23.15 to 24.96 in the 2005–2008 survey, and 23.43 to 25.28 in the 2013–2016 survey. Among women, BMI mean values ranged from 21.25 to 24.97 in the 1993–1996 survey, 21.72 to 25.30 in the 2005–2008 survey, and 22.26 to 24.73 in the 2013–2016 survey. We found there were increasing trends in the BMI mean values for most age groups in men except that the test for trend was marginally significant for men aged 50–59 ($p = 0.058$). However, the BMI mean values did not change much among survey waves for women across all age groups, as the tests for trend were not statistically significant.

Among people who attained high school or above, there were significant increasing trends in the BMI mean values for those aged 20–29 ($p<0.001$) and 30–39 ($p<0.001$). Among people with less than a high school degree, there were increasing trends in the BMI mean values for those aged 40–49 ($p = 0.008$), 60–69 ($p = 0.011$) and 70–79 ($p = 0.012$). People aged 20–29, 30–39, 50–59 and 60–69 in rural areas had significant increasing trends in the BMI mean values ($p<0.001$, $p = 0.028$, $p = 0.049$ and $p = 0.021$, respectively); in the urban areas, only people aged 30–39 had an increasing trend in the BMI mean values ($p = 0.007$).

The trends in the WC mean values appeared to be more consistently increasing than the trends in the BMI mean values among various defined subgroups. We found statistically

**Table 2. Trends in mean body mass index and waist circumference among Taiwanese adults: the NAHSIT 1993–2016.**

| | Body mass index (kg/m²) | | | | | | | Waist circumference (cm) | | | | | | |
|---|---|---|---|---|---|---|---|---|---|---|---|---|---|---|
| | 1993–1996 | | 2005–2008 | | 2013–2016 | | *p* for trend* | 1993–1996 | | 2005–2008 | | 2013–2016 | | *p* for trend* |
| | n | Mean (SE) | n | Mean (SE) | n | Mean (SE) | | n | Mean (SE) | n | Mean (SE) | n | Mean (SE) | |
| **Men** | | | | | | | | | | | | | | |
| 20–29 | 144 | 21.78 (0.34) | 137 | 23.15 (0.41) | 206 | 23.43 (0.48) | 0.002 | 143 | 73.54 (0.89) | 137 | 78.97 (1.19) | 168 | 81.20 (1.09) | <0.001 |
| 30–39 | 209 | 23.20 (0.30) | 121 | 24.73 (0.31) | 198 | 25.16 (0.43) | <0.001 | 209 | 78.84 (1.02) | 121 | 84.04 (0.87) | 167 | 86.70 (1.14) | <0.001 |
| 40–49 | 291 | 23.43 (0.26) | 221 | 24.46 (0.18) | 218 | 25.19 (0.31) | <0.001 | 291 | 80.60 (0.53) | 220 | 84.59 (0.52) | 182 | 87.76 (0.87) | <0.001 |
| 50–59 | 270 | 23.70 (0.54) | 239 | 24.96 (0.20) | 303 | 24.86 (0.21) | 0.058 | 270 | 82.85 (1.74) | 237 | 86.96 (0.56) | 282 | 88.35 (0.68) | 0.003 |
| 60–69 | 345 | 23.67 (0.28) | 229 | 24.68 (0.29) | 328 | 25.28 (0.26) | <0.001 | 344 | 83.69 (0.95) | 229 | 88.25 (0.84) | 340 | 90.77 (0.75) | <0.001 |
| 70–79 | 148 | 22.87 (0.57) | 264 | 24.01 (0.37) | 234 | 24.52 (0.30) | 0.013 | 148 | 81.42 (2.90) | 263 | 87.00 (1.21) | 236 | 90.34 (0.81) | 0.005 |
| **Women** | | | | | | | | | | | | | | |
| 20–29 | 184 | 21.25 (0.39) | 165 | 21.72 (0.37) | 170 | 22.26 (0.40) | 0.078 | 183 | 67.26 (1.03) | 165 | 72.02 (0.96) | 202 | 74.95 (0.96) | <0.001 |
| 30–39 | 291 | 22.51 (0.31) | 148 | 22.20 (0.37) | 168 | 23.44 (0.53) | 0.182 | 290 | 70.71 (0.55) | 148 | 73.93 (0.98) | 197 | 78.96 (1.30) | <0.001 |
| 40–49 | 322 | 23.72 (0.20) | 255 | 23.82 (0.38) | 185 | 23.17 (0.31) | 0.169 | 322 | 73.33 (0.34) | 255 | 76.72 (0.92) | 217 | 78.23 (0.88) | <0.001 |
| 50–59 | 339 | 24.77 (0.19) | 252 | 24.40 (0.29) | 285 | 24.13 (0.25) | 0.058 | 339 | 77.94 (0.47) | 250 | 79.18 (0.78) | 301 | 81.75 (0.66) | <0.001 |
| 60–69 | 299 | 24.97 (0.26) | 238 | 25.25 (0.29) | 344 | 24.40 (0.29) | 0.112 | 299 | 81.29 (0.75) | 238 | 83.27 (1.00) | 321 | 82.95 (0.81) | 0.159 |
| 70–79 | 147 | 24.88 (0.26) | 226 | 25.30 (0.41) | 241 | 24.73 (0.30) | 0.703 | 147 | 81.04 (0.80) | 223 | 86.61 (1.06) | 229 | 84.67 (0.69) | 0.001 |
| **HS or above** | | | | | | | | | | | | | | |
| 20–29 | 223 | 21.15 (0.25) | 267 | 22.45 (0.29) | 353 | 22.72 (0.30) | <0.001 | 221 | 69.14 (0.62) | 267 | 75.40 (0.73) | 347 | 77.58 (0.75) | <0.001 |
| 30–39 | 250 | 22.26 (0.21) | 223 | 23.47 (0.34) | 332 | 24.10 (0.35) | <0.001 | 249 | 73.18 (0.45) | 223 | 79.12 (0.94) | 330 | 82.14 (0.96) | <0.001 |
| 40–49 | 188 | 23.53 (0.22) | 291 | 23.94 (0.26) | 324 | 23.86 (0.18) | 0.355 | 188 | 77.57 (0.34) | 290 | 80.63 (0.75) | 322 | 82.00 (0.61) | <0.001 |
| 50–59 | 91 | 23.19 (0.37) | 213 | 24.24 (0.28) | 353 | 24.34 (0.28) | 0.198 | 91 | 78.82 (1.25) | 212 | 82.56 (0.96) | 352 | 84.83 (0.89) | 0.003 |
| 60–69 | 63 | 23.50 (0.58) | 88 | 24.75 (0.42) | 289 | 24.37 (0.29) | 0.611 | 63 | 81.95 (1.56) | 88 | 85.68 (1.38) | 285 | 85.54 (0.94) | 0.229 |
| 70–79 | 27 | 24.56 (1.01) | 80 | 24.27 (0.59) | 106 | 23.90 (0.56) | 0.541 | 27 | 76.73 (9.02) | 80 | 87.05 (1.38) | 105 | 87.03 (1.31) | 0.247 |
| **Less than HS** | | | | | | | | | | | | | | |
| 20–29 | 105 | 22.37 (0.60) | 35 | 21.47 (0.68) | 23 | 24.50 (0.99) | 0.666 | 105 | 73.66 (1.22) | 35 | 73.86 (1.93) | 23 | 82.99 (2.65) | 0.047 |
| 30–39 | 249 | 23.44 (0.62) | 46 | 23.62 (0.66) | 34 | 25.66 (1.00) | 0.182 | 249 | 76.38 (1.25) | 46 | 78.85 (1.84) | 34 | 86.17 (2.74) | 0.004 |
| 40–49 | 423 | 23.60 (0.26) | 185 | 24.58 (0.36) | 79 | 25.25 (0.75) | 0.008 | 423 | 76.73 (0.43) | 185 | 80.81 (1.00) | 77 | 85.43 (1.76) | <0.001 |
| 50–59 | 517 | 24.40 (0.39) | 278 | 25.07 (0.32) | 234 | 24.69 (0.20) | 0.403 | 517 | 80.61 (0.98) | 275 | 83.41 (0.83) | 230 | 84.82 (0.66) | 0.001 |
| 60–69 | 580 | 24.37 (0.23) | 378 | 25.06 (0.22) | 383 | 25.26 (0.30) | 0.011 | 579 | 82.64 (0.72) | 378 | 85.67 (0.67) | 376 | 87.81 (0.83) | <0.001 |
| 70–79 | 268 | 23.72 (0.37) | 409 | 24.73 (0.26) | 368 | 24.84 (0.22) | 0.012 | 268 | 81.82 (1.25) | 405 | 86.74 (0.71) | 359 | 87.51 (0.66) | <0.001 |
| **Urban** | | | | | | | | | | | | | | |
| 20–29 | 138 | 21.69 (0.31) | 131 | 22.31 (0.41) | 187 | 22.38 (0.36) | 0.113 | 138 | 70.65 (0.59) | 131 | 75.06 (0.90) | 183 | 76.94 (0.98) | <0.001 |
| 30–39 | 188 | 22.82 (0.32) | 132 | 23.36 (0.31) | 179 | 24.35 (0.45) | 0.007 | 188 | 74.86 (0.56) | 132 | 79.02 (0.79) | 177 | 82.68 (1.20) | <0.001 |
| 40–49 | 257 | 23.50 (0.22) | 240 | 23.88 (0.26) | 194 | 24.09 (0.26) | 0.075 | 257 | 77.17 (0.23) | 240 | 79.84 (0.70) | 193 | 82.58 (0.85) | <0.001 |
| 50–59 | 204 | 24.44 (0.36) | 227 | 24.23 (0.23) | 291 | 24.11 (0.26) | 0.462 | 204 | 80.70 (0.95) | 226 | 82.22 (0.78) | 289 | 84.31 (0.82) | 0.004 |
| 60–69 | 200 | 24.51 (0.14) | 226 | 25.02 (0.19) | 327 | 24.63 (0.31) | 0.770 | 200 | 83.41 (0.49) | 226 | 85.73 (0.82) | 321 | 86.10 (0.97) | 0.019 |
| 70–79 | 100 | 24.08 (0.27) | 199 | 24.65 (0.36) | 214 | 24.64 (0.32) | 0.159 | 100 | 81.46 (2.07) | 197 | 86.60 (1.05) | 209 | 87.88 (0.80) | 0.004 |
| **Rural** | | | | | | | | | | | | | | |
| 20–29 | 190 | 21.06 (0.48) | 171 | 22.53 (0.36) | 189 | 23.55 (0.45) | <0.001 | 188 | 70.11 (1.46) | 171 | 75.73 (1.27) | 187 | 79.33 (1.06) | <0.001 |
| 30–39 | 312 | 22.95 (0.29) | 137 | 23.78 (0.60) | 187 | 23.94 (0.33) | 0.028 | 311 | 74.57 (1.56) | 137 | 79.23 (1.71) | 187 | 81.96 (0.80) | <0.001 |
| 40–49 | 356 | 23.81 (0.24) | 236 | 24.69 (0.38) | 209 | 24.12 (0.30) | 0.483 | 356 | 76.58 (0.99) | 235 | 82.49 (0.70) | 206 | 82.55 (0.80) | <0.001 |
| 50–59 | 405 | 23.76 (0.53) | 264 | 25.52 (0.34) | 297 | 25.22 (0.22) | 0.049 | 405 | 79.57 (1.87) | 261 | 84.48 (1.15) | 294 | 85.87 (0.65) | 0.003 |
| 60–69 | 444 | 23.81 (0.53) | 241 | 24.89 (0.31) | 345 | 25.17 (0.24) | 0.021 | 443 | 80.89 (1.85) | 241 | 85.53 (1.05) | 340 | 87.76 (0.83) | 0.001 |
| 70–79 | 195 | 23.33 (0.59) | 291 | 24.63 (0.35) | 261 | 24.62 (0.30) | 0.073 | 195 | 80.85 (2.25) | 289 | 87.15 (0.96) | 256 | 86.71 (0.92) | 0.032 |

*Linear trends in the mean body mass index and waist circumference were tested using linear regression model.

HS = high school, NAHSIT = Nutrition and Health Survey in Taiwan, SE = standard error.

significant increasing trends in the WC mean values across all age groups stratified by sex, educational levels and urbanization levels. The test for trend was not significant in only three groups: women aged 60–69 ($p = 0.159$) and people aged 60–69 and 70–79 with high school or above education ($p = 0.229$ and $p = 0.247$, respectively).

Table 3 presents the trends in the prevalence of underweight and overweight for each age group stratified by sex, educational levels and urbanization levels. Overall, people aged 20–29 had the highest prevalence of underweight. The prevalence of underweight decreased from 1993–1996 to 2005–2008 but increased slightly from 2005–2008 to 2013–2016. We found statistically significant decreasing trends in the prevalence of underweight among people aged 60–69 except for the subgroup of women. People aged 20–29 residing in the rural areas and people aged 70–79 with lower than high school education or residing in the urban areas also had statistically significant decreasing trends ($p = 0.043$, $p = 0.023$ and $p = 0.033$, respectively).

The trends of overweight prevalence were increasing among men across all age groups except for people aged 20–29. But among women, the prevalence of overweight decreased significantly in those aged 40–49 and 50–59 ($p = 0.018$ and $p = 0.048$, respectively). Statistically significant increasing trends of overweight were found in several subgroups: people with high school or higher education aged 20–29 and 30–39, people with lower than high school education aged 60–69 and 70–79, people in the urban areas aged 30–39 and 70–79, and people in the rural areas aged 20–29, 50–59, and 60–69.

We present the trends in the prevalence of general obesity and abdominal obesity for each age group stratified by sex, educational levels and urbanization levels in Table 4. The prevalence of general obesity consistently increased among young and middle-aged populations but not among older populations without regard to the defined subgroups. For instance, significant increasing trends were observed among people aged 30–39 in all subgroups. The most rapid increase in the prevalence of general obesity was observed among those with lower than high school education aged 30–39 (9% to 35%, $p = 0.006$) and 40–49 (13% to 38%, $p = 0.003$).

The increasing trends in the prevalence of abdominal obesity were similar to the trends in WC mean values. We found statistically significant increasing trends in the prevalence of abdominal obesity across all age groups stratified by sex, educational levels and urbanization levels. The test for trend was not significant in only three groups: women aged 60–69 ($p = 0.135$) and people aged 20–29 with lower than high school education and residing in rural areas ($p = 0.631$ and $p = 0.091$, respectively). We observed the highest prevalence of abdominal obesity (77%) among women aged 70–79 in the 2005–2008 survey.

## Discussion

Our study aimed at examining the age-specific trends in the following: (1) distribution of BMI and WC and (2) prevalence of underweight, overweight, general obesity and abdominal obesity in Taiwan, stratified by sex, education, and urbanization. Our findings suggest that, although the overall trend of mean BMI and WC has levelled off, the prevalence of abdominal obesity has increased nearly two-fold from 1993 to 2016.

After conducting further age-specific analysis, the overall trends in the mean BMI and WC as well as in the prevalence of overweight, general obesity, and abdominal obesity could be briefly summarized as follows. A markedly increasing trend in the prevalence of general obesity was found among men; most results were not significant among women. Increasing trends were found particularly in younger- and middle-age adults with high school or higher education and in middle- and older-age adults with lower than high school education. Increasing trends were also found among people aged <39 and ≥50 years residing in rural areas and among those aged between 30 and 59 and ≥70 years residing in urban areas. We also found

**Table 3. Trends in prevalence of underweight and overweight among Taiwanese adults: the NAHSIT 1993–2016.**

| | Underweight | | | | | | | Overweight | | | | | | |
|---|---|---|---|---|---|---|---|---|---|---|---|---|---|---|
| | 1993–1996 | | 2005–2008 | | 2013–2016 | | *p* for trend* | 1993–1996 | | 2005–2008 | | 2013–2016 | | *p* for trend* |
| | n | % (SE) | n | % (SE) | n | % (SE) | | n | % (SE) | n | % (SE) | n | % (SE) | |
| **Men** | | | | | | | | | | | | | | |
| 20–29 | 144 | 17 (6) | 137 | 5 (2) | 170 | 10 (3) | 0.187 | 144 | 34 (5) | 137 | 47 (6) | 170 | 46 (5) | 0.061 |
| 30–39 | 209 | 3 (2) | 121 | 3 (2) | 168 | 4 (2) | 0.785 | 209 | 49 (5) | 121 | 69 (6) | 168 | 64 (4) | 0.009 |
| 40–49 | 291 | 5 (2) | 221 | 2 (1) | 185 | 1 (1) | 0.099 | 291 | 59 (5) | 221 | 70 (3) | 185 | 73 (4) | 0.039 |
| 50–59 | 270 | 5 (2) | 239 | 1 (1) | 285 | 2 (1) | 0.176 | 270 | 52 (8) | 239 | 73 (4) | 285 | 74 (4) | 0.012 |
| 60–69 | 345 | 5 (2) | 229 | 0 (0) | 344 | 0 (0) | <0.001 | 345 | 55 (2) | 229 | 69 (3) | 344 | 75 (3) | <0.001 |
| 70–79 | 148 | 10 (5) | 264 | 4 (1) | 241 | 2 (1) | 0.050 | 148 | 42 (8) | 264 | 59 (3) | 241 | 64 (4) | 0.020 |
| **Women** | | | | | | | | | | | | | | |
| 20–29 | 184 | 24 (4) | 165 | 18 (4) | 206 | 17 (3) | 0.152 | 184 | 21 (5) | 165 | 26 (5) | 206 | 31 (4) | 0.141 |
| 30–39 | 291 | 6 (3) | 148 | 6 (3) | 198 | 9 (4) | 0.589 | 291 | 31 (3) | 148 | 30 (5) | 198 | 37 (4) | 0.396 |
| 40–49 | 322 | 1 (1) | 255 | 5 (2) | 218 | 2 (1) | 0.333 | 322 | 59 (4) | 255 | 50 (5) | 218 | 45 (4) | 0.018 |
| 50–59 | 339 | 2 (1) | 252 | 1 (0) | 303 | 7 (2) | 0.061 | 339 | 68 (3) | 252 | 58 (5) | 303 | 58 (3) | 0.048 |
| 60–69 | 299 | 3 (1) | 238 | 2 (1) | 328 | 2 (1) | 0.389 | 299 | 67 (4) | 238 | 76 (3) | 328 | 58 (5) | 0.092 |
| 70–79 | 147 | 4 (2) | 226 | 3 (1) | 234 | 1 (1) | 0.095 | 147 | 66 (4) | 226 | 69 (4) | 234 | 65 (4) | 0.814 |
| **HS or above** | | | | | | | | | | | | | | |
| 20–29 | 223 | 21 (4) | 267 | 12 (2) | 353 | 14 (3) | 0.104 | 223 | 27 (3) | 267 | 37 (4) | 353 | 38 (3) | 0.006 |
| 30–39 | 250 | 6 (2) | 223 | 5 (2) | 332 | 7 (2) | 0.846 | 250 | 34 (3) | 223 | 51 (5) | 332 | 47 (3) | 0.010 |
| 40–49 | 188 | 1 (1) | 291 | 4 (2) | 324 | 2 (1) | 0.986 | 188 | 57 (6) | 291 | 59 (3) | 324 | 57 (3) | 0.903 |
| 50–59 | 91 | 10 (5) | 213 | 1 (1) | 353 | 5 (2) | 0.670 | 91 | 54 (9) | 213 | 62 (5) | 353 | 64 (4) | 0.458 |
| 60–69 | 63 | 4 (2) | 88 | 3 (3) | 289 | 1 (1) | 0.017 | 63 | 60 (10) | 88 | 64 (6) | 289 | 64 (4) | 0.836 |
| 70–79 | 27 | 4 (3) | 80 | 2 (1) | 106 | 2 (2) | 0.580 | 27 | 58 (13) | 80 | 68 (6) | 106 | 59 (6) | 0.906 |
| **Less than HS** | | | | | | | | | | | | | | |
| 20–29 | 105 | 20 (7) | 35 | 8 (8) | 23 | 0 (0) | 0.189 | 105 | 29 (7) | 35 | 18 (7) | 23 | 34 (14) | 0.645 |
| 30–39 | 249 | 3 (1) | 46 | 0 (0) | 34 | 2 (1) | 0.255 | 249 | 45 (6) | 46 | 41 (5) | 34 | 71 (10) | 0.251 |
| 40–49 | 423 | 4 (1) | 185 | 2 (1) | 79 | 2 (1) | 0.225 | 423 | 61 (3) | 185 | 62 (5) | 79 | 63 (7) | 0.736 |
| 50–59 | 517 | 3 (1) | 278 | 1 (1) | 234 | 3 (2) | 0.998 | 517 | 61 (4) | 278 | 68 (4) | 234 | 69 (3) | 0.111 |
| 60–69 | 580 | 4 (1) | 378 | 1 (0) | 383 | 1 (1) | 0.027 | 580 | 61 (3) | 378 | 75 (3) | 383 | 68 (4) | 0.048 |
| 70–79 | 268 | 8 (3) | 409 | 3 (1) | 368 | 2 (1) | 0.023 | 268 | 53 (6) | 409 | 63 (3) | 368 | 66 (3) | 0.044 |
| **Urban** | | | | | | | | | | | | | | |
| 20–29 | 138 | 18 (5) | 131 | 12 (3) | 187 | 14 (4) | 0.431 | 138 | 30 (3) | 131 | 34 (5) | 187 | 34 (4) | 0.457 |
| 30–39 | 188 | 5 (2) | 132 | 3 (2) | 179 | 8 (3) | 0.659 | 188 | 38 (3) | 132 | 49 (5) | 179 | 49 (4) | 0.017 |
| 40–49 | 257 | 4 (1) | 240 | 5 (2) | 194 | 1 (1) | 0.228 | 257 | 57 (3) | 240 | 58 (3) | 194 | 58 (4) | 0.921 |
| 50–59 | 204 | 5 (1) | 227 | 1 (1) | 291 | 6 (2) | 0.623 | 204 | 63 (3) | 227 | 62 (4) | 291 | 62 (3) | 0.832 |
| 60–69 | 200 | 4 (1) | 226 | 2 (1) | 327 | 1 (1) | 0.033 | 200 | 64 (4) | 226 | 72 (2) | 327 | 64 (4) | 0.957 |
| 70–79 | 100 | 6 (3) | 199 | 2 (1) | 214 | 1 (1) | 0.033 | 100 | 54 (4) | 199 | 62 (3) | 214 | 68 (4) | 0.029 |
| **Rural** | | | | | | | | | | | | | | |
| 20–29 | 190 | 27 (7) | 171 | 12 (5) | 189 | 12 (3) | 0.043 | 190 | 19 (8) | 171 | 39 (7) | 189 | 46 (4) | 0.016 |
| 30–39 | 312 | 2 (1) | 137 | 6 (3) | 187 | 4 (1) | 0.162 | 312 | 45 (4) | 137 | 51 (8) | 187 | 50 (4) | 0.360 |
| 40–49 | 356 | 1 (0) | 236 | 1 (1) | 209 | 3 (2) | 0.098 | 356 | 65 (5) | 236 | 64 (3) | 209 | 58 (3) | 0.171 |
| 50–59 | 405 | 0 (0) | 264 | 1 (1) | 297 | 2 (1) | 0.096 | 405 | 53 (8) | 264 | 72 (6) | 297 | 73 (3) | 0.042 |
| 60–69 | 444 | 3 (1) | 241 | 0 (0) | 345 | 0 (0) | 0.010 | 444 | 55 (2) | 241 | 72 (6) | 345 | 69 (4) | 0.004 |
| 70–79 | 195 | 9 (6) | 291 | 5 (1) | 261 | 2 (1) | 0.149 | 195 | 51 (13) | 291 | 67 (5) | 261 | 60 (4) | 0.545 |

*Linear trends in prevalence of underweight and overweight were tested using logistic regression model.

HS = high school, NAHSIT = Nutrition and Health Survey in Taiwan, SE = standard error.

**Table 4. Trends in prevalence of general and abdominal obesity among Taiwanese adults: the NAHSIT 1993–2016.**

| | General obesity | | | | | | | Abdominal obesity | | | | | | |
|---|---|---|---|---|---|---|---|---|---|---|---|---|---|---|
| | 1993–1996 | | 2005–2008 | | 2013–2016 | | *p* for trend* | 1993–1996 | | 2005–2008 | | 2013–2016 | | *p* for trend* |
| | n | % (SE) | n | % (SE) | n | % (SE) | | n | % (SE) | n | % (SE) | n | % (SE) | |
| **Men** | | | | | | | | | | | | | | |
| 20–29 | 144 | 5 (2) | 137 | 14 (3) | 170 | 13 (3) | 0.011 | 143 | 4 (2) | 137 | 17 (4) | 168 | 15 (3) | 0.003 |
| 30–39 | 209 | 8 (2) | 121 | 24 (5) | 168 | 27 (4) | <0.001 | 209 | 7 (3) | 121 | 27 (4) | 167 | 30 (4) | <0.001 |
| 40–49 | 291 | 10 (1) | 221 | 8 (2) | 185 | 25 (4) | 0.001 | 291 | 9 (2) | 220 | 18 (3) | 182 | 34 (4) | <0.001 |
| 50–59 | 270 | 10 (4) | 239 | 21 (3) | 285 | 20 (3) | 0.067 | 270 | 25 (6) | 237 | 34 (4) | 282 | 45 (4) | 0.01 |
| 60–69 | 345 | 15 (3) | 229 | 15 (4) | 344 | 20 (4) | 0.226 | 344 | 22 (3) | 229 | 43 (6) | 340 | 53 (4) | <0.001 |
| 70–79 | 148 | 12 (4) | 264 | 14 (4) | 241 | 17 (3) | 0.313 | 148 | 25 (8) | 263 | 39 (5) | 236 | 52 (4) | 0.006 |
| **Women** | | | | | | | | | | | | | | |
| 20–29 | 184 | 7 (3) | 165 | 10 (2) | 206 | 13 (3) | 0.204 | 183 | 7 (3) | 165 | 16 (3) | 202 | 22 (4) | 0.014 |
| 30–39 | 291 | 5 (2) | 148 | 10 (3) | 198 | 20 (4) | 0.003 | 290 | 9 (2) | 148 | 20 (4) | 197 | 36 (5) | <0.001 |
| 40–49 | 322 | 12 (2) | 255 | 16 (4) | 218 | 12 (2) | 0.931 | 322 | 18 (3) | 255 | 35 (4) | 217 | 42 (5) | <0.001 |
| 50–59 | 339 | 22 (2) | 252 | 18 (3) | 303 | 19 (3) | 0.605 | 339 | 38 (2) | 250 | 46 (4) | 301 | 55 (3) | <0.001 |
| 60–69 | 299 | 22 (4) | 238 | 25 (4) | 328 | 18 (3) | 0.427 | 299 | 52 (5) | 238 | 61 (5) | 321 | 62 (4) | 0.135 |
| 70–79 | 147 | 25 (4) | 226 | 28 (4) | 234 | 15 (3) | 0.079 | 147 | 52 (5) | 223 | 77 (4) | 229 | 76 (4) | <0.001 |
| **HS or above** | | | | | | | | | | | | | | |
| 20–29 | 223 | 3 (2) | 267 | 12 (2) | 353 | 13 (2) | 0.001 | 221 | 3 (2) | 267 | 16 (3) | 347 | 19 (3) | <0.001 |
| 30–39 | 250 | 4 (2) | 223 | 17 (3) | 332 | 22 (3) | <0.001 | 249 | 3 (2) | 223 | 23 (3) | 330 | 31 (3) | <0.001 |
| 40–49 | 188 | 8 (1) | 291 | 10 (2) | 324 | 13 (2) | 0.039 | 188 | 8 (2) | 290 | 24 (3) | 322 | 36 (4) | <0.001 |
| 50–59 | 91 | 6 (3) | 213 | 15 (3) | 353 | 17 (3) | 0.267 | 91 | 20 (6) | 212 | 33 (4) | 352 | 50 (4) | <0.001 |
| 60–69 | 63 | 8 (4) | 88 | 16 (5) | 289 | 15 (3) | 0.539 | 63 | 26 (5) | 88 | 36 (5) | 285 | 50 (4) | 0.001 |
| 70–79 | 27 | 24 (14) | 80 | 10 (5) | 106 | 9 (4) | 0.237 | 27 | 27 (9) | 80 | 44 (7) | 105 | 48 (6) | 0.095 |
| **Less than HS** | | | | | | | | | | | | | | |
| 20–29 | 105 | 11 (6) | 35 | 14 (6) | 23 | 12 (6) | 0.862 | 105 | 12 (6) | 35 | 16 (6) | 23 | 14 (7) | 0.631 |
| 30–39 | 249 | 9 (3) | 46 | 19 (7) | 34 | 35 (13) | 0.006 | 249 | 13 (4) | 46 | 27 (6) | 34 | 53 (13) | 0.002 |
| 40–49 | 423 | 13 (2) | 185 | 16 (4) | 79 | 38 (8) | 0.003 | 423 | 16 (3) | 185 | 32 (5) | 77 | 49 (8) | <0.001 |
| 50–59 | 517 | 18 (3) | 278 | 23 (3) | 234 | 25 (3) | 0.092 | 517 | 34 (3) | 275 | 47 (4) | 230 | 50 (4) | 0.001 |
| 60–69 | 580 | 19 (2) | 378 | 21 (3) | 383 | 24 (3) | 0.256 | 579 | 37 (3) | 378 | 57 (4) | 376 | 64 (3) | <0.001 |
| 70–79 | 268 | 17 (2) | 409 | 24 (3) | 368 | 19 (3) | 0.497 | 268 | 39 (6) | 405 | 61 (4) | 359 | 69 (3) | <0.001 |
| **Urban** | | | | | | | | | | | | | | |
| 20–29 | 138 | 5 (2) | 131 | 13 (3) | 187 | 11 (3) | 0.033 | 138 | 5 (2) | 131 | 16 (2) | 183 | 15 (4) | 0.007 |
| 30–39 | 188 | 7 (1) | 132 | 15 (3) | 179 | 24 (4) | <0.001 | 188 | 8 (2) | 132 | 22 (3) | 177 | 33 (4) | <0.001 |
| 40–49 | 257 | 12 (1) | 240 | 11 (3) | 194 | 16 (3) | 0.260 | 257 | 14 (2) | 240 | 26 (3) | 193 | 39 (5) | <0.001 |
| 50–59 | 204 | 18 (3) | 227 | 15 (2) | 291 | 18 (3) | 0.954 | 204 | 34 (3) | 226 | 36 (4) | 289 | 49 (4) | 0.003 |
| 60–69 | 200 | 20 (2) | 226 | 22 (2) | 327 | 18 (4) | 0.719 | 200 | 39 (3) | 226 | 52 (5) | 321 | 58 (5) | 0.001 |
| 70–79 | 100 | 19 (4) | 199 | 22 (5) | 214 | 16 (3) | 0.684 | 100 | 41 (4) | 197 | 56 (5) | 209 | 67 (4) | <0.001 |
| **Rural** | | | | | | | | | | | | | | |
| 20–29 | 190 | 8 (6) | 171 | 9 (2) | 189 | 17 (3) | 0.343 | 188 | 8 (6) | 171 | 17 (5) | 187 | 24 (3) | 0.091 |
| 30–39 | 312 | 7 (4) | 137 | 23 (7) | 187 | 20 (4) | 0.046 | 311 | 8 (3) | 137 | 27 (5) | 187 | 33 (5) | <0.001 |
| 40–49 | 356 | 6 (1) | 236 | 15 (3) | 209 | 20 (3) | 0.001 | 356 | 10 (3) | 235 | 27 (4) | 206 | 37 (4) | <0.001 |
| 50–59 | 405 | 11 (13) | 264 | 28 (3) | 297 | 24 (3) | 0.044 | 405 | 26 (8) | 261 | 47 (4) | 294 | 54 (4) | 0.006 |
| 60–69 | 444 | 15 (16) | 241 | 16 (4) | 345 | 21 (3) | 0.329 | 443 | 30 (9) | 241 | 52 (5) | 340 | 56 (3) | 0.011 |
| 70–79 | 195 | 16 (2) | 291 | 20 (3) | 261 | 17 (3) | 0.949 | 195 | 31 (8) | 289 | 60 (5) | 256 | 61 (4) | 0.007 |

*Linear trends in prevalence of general and abdominal obesity were tested using logistic regression model.

HS = high school, NAHSIT = Nutrition and Health Survey in Taiwan, SE = standard error.

that there is a decreasing trend in the prevalence of underweight, particularly among those aged 60–79 years.

The Non-Communicable Disease Risk Factor Collaboration (NCD-RisC) reported that the overall prevalence of general obesity continues to be high and estimated that the prevalence is expected to increase up to 18% in men and 21% in women worldwide by 2025[14]. In addition, the prevalence of abdominal obesity in the US has increased steadily up to more than 50% since 2004 [15]. Our findings are consistent with studies that indicate the overall trend of general and abdominal obesity has been increasing progressively over time [16,17].

Studies have also indicated that the prevalence of general and abdominal obesity was generally higher among women than among men, regardless of age [17]. We had similar patterns in the 1993–1996 and the 2005–2008 surveys. In the 2013–2016 survey, however, the prevalence of general obesity among women was the same or lower than among men for most age groups. The overall prevalence of abdominal obesity has increased over time regardless of sex in countries such as the US and China, but the findings in Korea showed a decreasing trend among women from 1998 to 2014 [6,8]. In our study, the prevalence of abdominal obesity increased rapidly regardless of sex, yet the upward trend was flattening among women aged $\geq$ 60 years. Although women in most populations worldwide are more likely to be obese than men due to a variety of sociocultural and socioeconomic reasons and their reproductive role [18,19], such a trend was not found in the recent (2013–2016) survey in Taiwan. Studies have highlighted how social norms and thin-ideal media exposure may influence the obesity prevalence, especially among women [20,21]. Women in East Asian countries such as Taiwan and South Korea were found to be more prone to the societal pressures of thin-ideal body image [22]. There is an increasing trend of media consumption and internet use in Taiwan, and internet users have increased from about 9.5 million in 2003 to 18.8 million in 2016 [23–25]. Women are at a greater risk of being exposed to the thin-ideal image and social pressure to be thin, resulting in the increase of body dissatisfaction, misperception of the ideal body, disordered eating behaviors, and unhealthy weight control, especially among female adolescents [22,26,27].

The impact of educational levels on obesity was inconsistent in previous studies. Studies have underscored that the differential measurements of educational attainment could capture different underlying constructs, such as years of education, degree of education, intelligence, and illiteracy or not, which could influence the association between educational attainment and health outcomes [28–31]. A systematic review on educational attainment and obesity concluded that a positive relationship between obesity and educational level was seen more in low-income countries, whereas a reverse relationship was more common in high-income countries [6,30,32,33]. The relationship can also be moderated by sex, as among women; the association between educational attainment and obesity is more consistent and further away from the null than among men, for which it has more variations [30]. Our results showed that the prevalence of general obesity and abdominal obesity among people with a lower than high school education was higher than people with a high school or higher education. This relationship between education and obesity may be affected by how the educational attainment was measured and other factors such as poverty levels and occupations [30,34–36], which requires further investigation.

Studies have also reported differed trends of general obesity and abdominal obesity stratified by the residential areas. The results from the National Nutrition Survey 1976–95 in Japan showed that regardless of age and sex, except for women aged 50 years and above, the prevalence of general obesity was generally higher in small towns than in cities or in metropolitan areas [37]. Similar patterns were was found in the US, that the obesity (BMI$\geq$30) and severe obesity (BMI $\geq$40) prevalence was higher in non-metropolitan statistical areas (MSAs; large: $\geq$1 million population) and in medium or small MSAs than in large MSAs during 2001 to

2016, regardless of sex [38]. However, the findings from China showed that people who resided in urban areas had a higher prevalence of general obesity from 1993 to 2008 than people in rural areas, but the prevalence was higher in the rural areas than the urban areas from 2009 to 2015 [6,39]. People residing in the urban areas also had a higher prevalence of abdominal obesity than those in rural areas, although the differences have decreased [6,39]. Our results were consistent with previous studies [6,37,39]. General obesity was more prevalent in rural areas, particularly in the 2013–2016 survey, while abdominal obesity was more prevalent only in rural areas in the 2005–2008 survey. Our results also suggested that general and abdominal obesity prevalence increased dramatically from 2008 to 2013 in rural areas and were in line with studies that examined the trends worldwide and in other countries [37,38,40]. The BMI findings in rural areas were persistently higher than urban areas in high-income and industrialized countries [40]. The increased prevalence of mechanized work in rural areas may result in decreased levels of physical activities and further result in an increased risk of obesity [41]. In addition, the lack of nutrition knowledge and the attitudes in rural areas may increase the likelihood of malnutrition or excessive consumption of low-quality calories, and could lead to increased prevalence of general and abdominal obesity [40,42].

This study has several limitations. First, we were not able to examine BMI and WC trajectories at the individual level because the data were from cross-sectional surveys instead of longitudinal studies. Second, although the multi-staged stratified and clustered sampling approach was used to collect data, selection bias could not be avoided. People who did not show up for the health examination and who had missing anthropometric data were not included in this study. Third, we only included people aged 20 years and above in our analysis. Since the general pattern of BMI and WC changes for children and adolescents may be quite different from adults, further studies are needed if our research question is also targeting younger populations. Last, factors such as body fat, muscle mass, caloric intake, and physical activities were not discussed in this study. The trend of BMI is strongly associated with body fat and muscle mass [43]. As we were not able to distinguish whether the high prevalence of general obesity was the result of high body fat or high muscle mass, there may be misclassification of being "fat," especially among men [43].

Despite the limitations, our study has some strengths. To our knowledge, this is one of the first studies in Taiwan to examine multiple waves of nationally representative data—NAHSIT 1993–1996, 2005–2008, and 2013–2016—to understand the pattern of WC and BMI. Instead of using the self-reported weight and height, which could be biased as people tend to overreport their height and underreport their weight, we used data collected by standardized equipment and techniques [44]. Based on the standardized measurements, our results provided a more precise estimation of the prevalence of underweight, overweight, general and abdominal obesity. Most importantly, our study also provided detailed information regarding the stratified age-specific prevalence of abdominal obesity, which has been widely reported to be associated with all-cause cardiovascular disease, and cancer mortality, even among people with normal BMI [45–47].

## Conclusions

The three waves of nationally representative surveys of adults in Taiwan demonstrated that the pattern of general and abdominal obesity varied by subgroup populations. Although the general obesity prevalence has decreased in certain age groups among women, the prevalence of abdominal obesity was increasing, indicating that women with normal BMI may have abdominal obesity. Besides, the WC increased rapidly over time, and the estimates for most of the age-specific prevalence of abdominal obesity also increased regardless of sex, educational

attainment, and urbanization levels. The results of our study emphasized the importance of developing and executing interventions to flatten and decrease the general and abdominal obesity prevalence and informed regarding which subgroup should be the future research target.

## Author Contributions

**Conceptualization:** Tzu-Jung Wong, Tsung Yu.

**Data curation:** Tzu-Jung Wong, Tsung Yu.

**Formal analysis:** Tzu-Jung Wong, Tsung Yu.

**Funding acquisition:** Tsung Yu.

**Supervision:** Tsung Yu.

**Writing – original draft:** Tzu-Jung Wong, Tsung Yu.

**Writing – review & editing:** Tzu-Jung Wong, Tsung Yu.

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
