## [Decision Letter · Decision Letter 0]

3 Jun 2022

PONE-D-22-05300Trends in the distribution of body mass index and waist circumference among Taiwanese adults, 1993-2016PLOS ONE

Dear Dr. Yu,

Thank you for submitting your manuscript to PLOS ONE. After careful consideration, we feel that it has merit but does not fully meet PLOS ONE’s publication criteria as it currently stands. Therefore, we invite you to submit a revised version of the manuscript that addresses the points raised during the review process.

We look forward to receiving your revised manuscript.

Kind regards,

Hugo Miguel Borges Sarmento

Academic Editor

PLOS ONE

Journal Requirements:

3.We note that the grant information you provided in the ‘Funding Information’ and ‘Financial Disclosure’ sections do not match. 

Reviewers' comments:

Reviewer's Responses to Questions

**Comments to the Author**

1. Is the manuscript technically sound, and do the data support the conclusions?

Reviewer #1: Partly

Reviewer #2: Yes

2. Has the statistical analysis been performed appropriately and rigorously? 

Reviewer #1: No

Reviewer #2: Yes

3. Have the authors made all data underlying the findings in their manuscript fully available?

Reviewer #1: Yes

Reviewer #2: Yes

4. Is the manuscript presented in an intelligible fashion and written in standard English?

Reviewer #1: Yes

Reviewer #2: Yes

5. Review Comments to the Author

Reviewer #1: The manuscript is suitable with the Plos One article type. However, minor changes should be performed in order to fit all the guidelines. Although the aim, authors refer to obesity prevalence it does not have an impact on the title that is used in the article. Keywords should be clarified and focusing the content and study aim (they seem general and not specific for the topic).

Definition for secular trend should appear in the introduction because it’s a term repeated on the document.

Which reference is used for general obesity and abdominal obesity? And why sometimes is used just obesity and others general obesity?

At methods, Stata 15 reference seems incomplete.

Reviewer #2: PLOS ONE

Submission: PONE-D-22-05300

An increase in mean body mass index or prevalence of obesity may be accompanied by changes in the population BMI distribution. This study aimed to examine how the distributions of BMI and waist circumference have changed in Taiwan over a 23-year interval (1998-2014).

I highlight:

1. The number of included data;

2. Study relevance;

3. Overall english.

Some details should be improved:

1. Data analysis;

2. Methods details.

I suggest authors examine the distributional changes (over time) in BMI (or WC), using square regressions for all individuals and subgroups (stratified by sex, age, educational level, and residence area), including graphs or figures in the manuscript according to previous related works:

Kim, S., Subramanian, S. V., Oh, J., & Razak, F. (2018). Trends in the distribution of body mass index and waist circumference among South Korean adults, 1998-2014. Eur J Clin Nutr, 72(2), 198-206. https://doi.org/10.1038/s41430-017-0024-7

6. PLOS authors have the option to publish the peer review history of their article (what does this mean?). If published, this will include your full peer review and any attached files.

Reviewer #1: No

Reviewer #2: No

---

## [Author Response · Author response to Decision Letter 0]

27 Jul 2022

Response to Reviewers

1. Reviewer #1: The manuscript is suitable with the Plos One article type. However, minor changes should be performed in order to fit all the guidelines. Although the aim, authors refer to obesity prevalence it does not have an impact on the title that is used in the article. Keywords should be clarified and focusing the content and study aim (they seem general and not specific for the topic).

Response:

 We made changes to the title (page 1 in the manuscript with tracked changes) and keywords (page 4) accordingly. 

2. Definition for secular trend should appear in the introduction because it’s a term repeated on the document.

Response:

 In this study secular trend is the long-term trend, and we made changes to the Introduction section accordingly (line 20 on page 6).

3. Which reference is used for general obesity and abdominal obesity? And why sometimes is used just obesity and others general obesity?

Response:

 We updated reference 1 (N Engl J Med. 2008 Nov 13;359(20):2105-20), where there are definitions for general and abdominal obesity. We clarified these two terms throughout the manuscript.

4. At methods, Stata 15 reference seems incomplete.

Response:

 We edited the reference accordingly (line 92 on page 9).

5. Reviewer #2: PLOS ONE

Submission: PONE-D-22-05300

An increase in mean body mass index or prevalence of obesity may be accompanied by changes in the population BMI distribution. This study aimed to examine how the distributions of BMI and waist circumference have changed in Taiwan over a 23-year interval (1998-2014).

I highlight:

1. The number of included data;

2. Study relevance;

3. Overall english.

Response:

 We are not fully clear about where we should revise specifically. The number of data can be found in the Methods section (line 60 on page 8). Study relevance are clarified in the Introduction section. We had the manuscript edited by a professional English editor. 

6. Some details should be improved:

1. Data analysis;

2. Methods details.

Response:

 We edited the Methods section accordingly to clarify the data analysis.

7. I suggest authors examine the distributional changes (over time) in BMI (or WC), using square regressions for all individuals and subgroups (stratified by sex, age, educational level, and residence area), including graphs or figures in the manuscript according to previous related works:

Kim, S., Subramanian, S. V., Oh, J., & Razak, F. (2018). Trends in the distribution of body mass index and waist circumference among South Korean adults, 1998-2014. Eur J Clin Nutr, 72(2), 198-206. https://doi.org/10.1038/s41430-017-0024-7

Response:

 We thank for the suggestions. We in fact did use linear regression and logistic regression when we were testing for the trend effects (see lines 85-91 on page 9).

---

## [Editor Report · Decision Letter 1]

23 Aug 2022

Trends in the distribution of body mass index, waist circumference and prevalence of obesity among Taiwanese adults, 1993-2016

PONE-D-22-05300R1

Dear Dr. Tsung Yu,

We’re pleased to inform you that your manuscript has been judged scientifically suitable for publication and will be formally accepted for publication once it meets all outstanding technical requirements.

Kind regards,

Hugo Miguel Borges Sarmento

Academic Editor

PLOS ONE
---

## [Editor Report · Acceptance letter]

1 Sep 2022

PONE-D-22-05300R1 

Trends in the distribution of body mass index, waist circumference and prevalence of obesity among Taiwanese adults, 1993-2016 

Dear Dr. Yu:

I'm pleased to inform you that your manuscript has been deemed suitable for publication in PLOS ONE. Congratulations! Your manuscript is now with our production department. 

Kind regards, 

on behalf of

Dr. Hugo Miguel Borges Sarmento 

Academic Editor

PLOS ONE